# Delineating the Molecular Events Underlying Development of Prostate Cancer Variants with Neuroendocrine/Small Cell Carcinoma Characteristics

**DOI:** 10.3390/ijms222312742

**Published:** 2021-11-25

**Authors:** Mayuko Kanayama, Jun Luo

**Affiliations:** Department of Urology, James Buchanan Brady Urological Institute, Johns Hopkins University School of Medicine, Baltimore, MD 21205, USA; jluo2@jhmi.edu

**Keywords:** lineage plasticity, neuroendocrine/small cell carcinoma, aggressive variant prostate cancer

## Abstract

The treatment landscape of prostate cancer has changed dramatically following the advent of novel systemic therapies, most of which target the androgen receptor (AR). Agents such as abiraterone, enzalutamide, apalutamide, darolutamide were designed to further suppress androgen receptor signaling following gonadal suppression achieved by first-line androgen deprivation therapies. These potent AR targeting agents are increasingly used in the earlier stages of the disease spectrum with the goal of delaying disease progression and extending survival. Although these therapies are effective in controlling prostate tumors dependent on or addicted to AR signaling, prostate tumors surviving the onslaught of potent treatments may evolve and develop drug resistance. A substantial proportion of treatment failures can be explained by the development of treatment-induced aggressive prostate cancer variants such as neuroendocrine/small cell carcinoma. These emerging disease entities demand detailed characterization and precise definitions. We postulate that these treatment-induced prostate cancer entities should be defined molecularly to overcome the drawbacks associated with the current clinical and pathological definitions. A precise molecular definition conforms with current knowledge on the molecular evolution of this disease entity and will enable early detection and early intervention.

## 1. Introduction

Most treatment-naïve prostate cancers (PCa) depend on androgen receptor (AR) signaling to proliferate. Therefore, androgen deprivation therapy (ADT) (e.g., luteinizing hormone agonist/antagonist with or without casodex) is historically the first choice of treatment for advanced castration-sensitive PCa (CSPC). However, treated patients eventually develop resistance to first-line ADT and progress to castration-resistant PCa (CRPC). A subset of CRPC may progress to lethal AR-indifferent neuroendocrine/small cell carcinoma (NE/SC) following further AR suppression by the second and third-generation AR inhibitors (e.g., abiraterone, enzalutamide, apalutamide, darolutamide). A recent study showed that up to 17% of CRPC patients undergoing treatment with abiraterone/enzalutamide have histologically discernable NE/SC lesions on biopsies [1]. Emerging data suggest that PCa utilizes lineage plasticity to their advantage to evade selective pressure imposed by these potent AR-targeting agents [2]. Lineage plasticity generally describes the ability of living organisms to express variable phenotypes depending on surrounding environments. In this context, lineage plasticity involves phenotypic conversion that enables prostate cancer cells to survive in androgen-deprived conditions and obtain NE/SC characteristics.

While there have been efforts in standardizing histological diagnostic categorization for *de novo* as well as treatment-induced prostate cancer variants with NE/SC characteristics [3], some CRPC tumors that lost AR dependence may not show typical morphological and immunohistochemical characteristics. In addition, histology-based definition does not exist for CRPC tumors in a transition state to be “full-fledged” NE/SC, primarily due to histological continuum characterized by the mixed and overlapping morphologies and intra-patient and intra-tumor heterogeneity. Lineage plasticity may also compromise the ability to define these tumors. Yet, there is a clinical need to definitively characterize the subset of CRPC in the transition state to NE/SC due to their treatment implications. Identification of prostate tumors in transition to NE/SC will enable early detection and early intervention. From clinical perspective, the concept of “Aggressive variant prostate cancer” (AVPC) has been proposed to enable early detection [4,5]. One potential drawback of this AVPC criteria is that AVPC cannot be defined until relatively late stages of the disease unless patients have histologically confirmed NE/SC. Given these drawbacks of the pathological and clinical definitions, one may wonder whether detection of the molecular drivers that may occur prior to pathological and clinical manifestations could further refine existing definitions, so that NE/SC or other resistance phenotypes can be captured/detected in the transition state rather than at the end of the spectrum. In this review, we aim to summarize temporal delineation of the molecular events during the NE/SC development and discuss the potential of utilizing molecular definition to complement existing pathological and clinical definitions. This collective definition will allow precision therapy and may drive efforts in early detection (noninvasive methods, or imaging methods taking advantage of NE/SC markers) and early escalation treatments.

## 2. Pathological Definition of Prostate Cancer Variants with NE/SC Characteristics

We would like to first visit the existing definitions of prostate cancer variants with NE/SC characteristics. According to the 2016 WHO (World Health Organization) Classification, prostate tumors with neuroendocrine differentiation (NED) are classified into the following four categories: adenocarcinoma with NED, well-differentiated neuroendocrine tumor (carcinoid), small cell neuroendocrine carcinoma and large cell neuroendocrine carcinoma [6]. However, another version of classification has been proposed by Epstein et al. considering the unique aspects of NED in PCa. Based on their classification, prostate neuroendocrine tumors fall into the following six categories: (i) usual prostate adenocarcinoma with NED; (ii) adenocarcinoma with Paneth cell NED; (iii) carcinoid tumor; (iv) small cell carcinoma; (v) large cell neuroendocrine carcinoma; and (vi) mixed (small or large cell) neuroendocrine carcinoma-acinar adenocarcinoma [3] (Table 1). Among them, (i) and (ii) are molecularly distinct from (iv) in that (i) and (ii) did not differ by prevalence of *TP53* missense mutations, or *PTEN* or *RB1* loss when compared with those without NED, as recently reported by Kaur et al. [7]. Furthemore, (i) and (ii) are less likely to be associated with adverse clinical outcomes [7,8]. Thus, although further studies are needed, (i) and (ii) probably should be considered as the different disease entity from (iv) which is associated with aggressive phenotypes. In addition, (iii) and (v) are extremely rare variants and yet to be fully characterized. Therefore, to avoid confusion, the term “NE/SC” used in this review will specifically refer to (iv) small cell carcinoma or components of small cell carcinoma in (vi), whereas NEPCa (neuroendocrine prostate cancer) collectively refers to prostate cancers with NED listed in Table 1. In addition, we kept the original terminology in cited papers, although many preclinical studies did not involve diagnostic definitions.

Table 2 summarizes a set of IHC markers used for NEPCa diagnosis, some of which are exploratory markers such as Cyclin D1, YAP1, and FoxA2. NEPCa usually does not express markers generally specific to prostate cells (e.g., PSA), but characteristically expresses markers of neural lineage such as chromogranin A, synaptophysin, TTF-1 and CD56 (Table 2). Using a combination of markers improves diagnostic accuracy of NEPCa [9,10]. It is worth emphasizing that not all CRPC that lost AR dependence can be identified by morphology in conjunction with IHC. Other morphologies in the transition state are difficult to evaluate due to histological continuum characterized by the mixed and overlapping morphologies and intra-patient/intra-tumor heterogeneity. Characterization of these transitional morphologies will be discussed in Section 6 of this review.

**Table 1 ijms-22-12742-t001:** The classification of neuroendocrine prostate cancer proposed by Epstein et al. [3] and WHO [6].

Epstein’s Classification	2016 WHO Classification	Description
(i) Usual prostate adenocarcinoma with NE differentiation	Adenocarcinoma with neuroendocrine differentiation	Morphologically typical acinar or ductal prostate adenocarcinoma with NED only demonstrated by IHC. This type of tumor is molecularly and clinically distinct form (iv) small cell carcinoma and is not associated with poor outcomes [7].
(ii) Adenocarcinoma with Paneth cell NE differentiation	Not defined	Histologically typical prostate adenocarcinoma containing varying proportions of cells with prominent eosinophilic cytoplasmic granules that are chromogranin positive and contain neurosecretory granules. Similar to (i), this type of tumor is molecularly distinct from (iv) [7], and is not associated with poor outcomes [8].
(iii) Carcinoid tumor	Well-differentiated neuroendocrine tumor	Well-differentiated NEPCa not closely associated with usual PCa. which are positive for NE markers and negative for PSA. This type of NEPCa is extremely rare and only reported in a limited number of case reports.
(iv) Small cell carcinoma	Small cell neuroendocrine carcinoma	The most well-studied aggressive NEPCa variant that usually arises under selective pressure of ADT (can arise de novo but rare). Defined by characteristic nuclear features, including lack of prominent nucleoli, nuclear molding, fragility, and crush artifact. High N/C ratio, indistinct cell borders, a high mitotic rate and apoptotic bodies are common.
(v) Large cell NE carcinoma	Large cell neuroendocrine carcinoma	An extremely rare NEPCa variant characterized by large nests with peripheral palisading and often geographic necrosis, prominent nucleoli, vesicular clumpy chromatin, and/or large cell size and abundant cytoplasm, a high mitotic rate. Positive for at least one NE marker by IHC. The largest series of seven cases was reported in 2006 [11].
(vi) Mixed (small or large cell) NE carcinoma-acinar adenocarcinoma	Not defined	Biphasic carcinoma with admixed components of NE (small cell or large cell) carcinoma and usual conventional acinar adenocarcinoma. This type of NEPCa is associated with high-grade aggressive disease. Less frequently, it shows overlap between (iv) small cell carcinoma and adenocarcinoma and is considered to be in the process of transdifferentiation.

Abbreviations: WHO—World Health Organization, NE—neuroendocrine, NED—neuroendocrine differentiation, IHC—immunohistochemistry, NEPCa—neuroendocrine prostate cancer, N/C—nuclear to cytoplasmic.

**Table 2 ijms-22-12742-t002:** A list of IHC markers used for NEPCa diagnosis.

	Marker	Description
Negative/low in NEPCa	PSA	PSA expression is positive throughout disease progression from CSPC to CRPC [12], but positivity decreases in NE/SC [9,10,12]. Yet, a subset of NE/SC (19%) is positive PSA [10].
	AR	AR transcriptional activity is low in NE/SC [1]. AR “null” mCRPC is enriched with *TP53*, *RB1* and *PTEN* alterations [13].
	Nkx 3.1	Nkx 3.1 is a highly sensitive and specific prostate adenocarcinoma marker [14], and has recently been the most frequently used prostate marker.
	PSAP	PSAP expression is positively correlated with PSA expression [15].
	P501s (prostein)	P501s positivity in NE/SC is 28% [10]. P501s is useful in identifying the prostatic origin of NE/SC than PSA [10].
	Cyclin D1	Cyclin D1 loss was observed in 88% of NE/SC and its loss was highly correlated with Rb loss [16].
	YAP1	YAP1 is increased in high-grade adeno PCa, but downregulated in NEPCa. Downregulation of YAP1 in NEPCa has been shown in several datasets [17].
Positive in NEPCa	Chromogranin A	Secretory granules produced by a variety of neural cells [18]. It can be used as a serum marker, too [19]. More than 60% of NEPCa is reported to be positive for CGA [9,10].
	Synaptophysin	A vesicle membrane protein that localizes in a variety of neural cells [20]. More than 80% of NEPCa is reported to be positive for SYP [9,10].
	CD56 (Neural cell adhesion molecule)	Membrane-bound glycoprotein predominantly expressed in neural cells. Although positivity in NEPCa is high [9,10], its specificity is low [21].
	TTF-1 (Thyroid transcription factor-1)	TTF-1 is a highly sensitive marker for extrapulmonary small cell carcinoma including NE/SC [9,22].
	FoxA2	A transcription factor specifically upregulated in NE/SC. Its positivity is reportedly higher than CGA or SYP in NE/SC [23].
	INSM1 (Insulinoma-associated protein 1)	Zinc-finger transcriptional factor elevated in NE/SC [24]. INSM1 is reported to be superior to CGA, SYP and CD56 [24,25].
	Ki67	A well-known marker of proliferation. Ki67 is >50–80% in NE/SC and LC NE carcinoma but usually not increased in other tumor types such as adenocarcinoma with Paneth cell NED and carcinoid tumor [3].

Abbreviations: PSA—prostate-specific antigen, AR—androgen receptor, mCRPC—metastatic castration-resistant prostate cancer, PSAP—prostatic acid phosphatase, CGA—Chromogranin A, SYP—Synaptophysin, LC NE carcinoma—large cell NE carcinoma.

## 3. Clinical Characteristics of Prostate Cancer Variants with NE/SC Characteristics

Clinical characteristics of NE/SC are quite different from that of CRPC adenocarcinoma, and they are often exemplified by low serum PSA, high serum NE markers (e.g., chromogranin A, neuron-specific enolase), visceral metastasis, and poor response to AR-targeted therapies [26]. Traditionally, NE/SC would be clinically suspected in patients presenting with rapidly progressive disease [27]. Recently, Aparicio et al. proposed an innovative concept of aggressive variant prostate carcinoma (AVPC) as inclusion criteria for platinum-based chemotherapy trials [4] (Table 3). Indeed, patients with AVPC showed a high response rate to platinum-containing chemotherapies [4]. Furthermore, in their subsequent study, clinically defined AVPC has been proven to share molecular features with NE/SC such as *RB1* loss and *TP53* loss-of-function mutations [28]. Together, AVPC criteria enabled identification of patients with clinical features of NE/SC without biopsies and is currently utilized for patients selection in clinical trials [29]. However, like pathological definitions, one potential drawback of the AVPC criteria is that AVPC cannot be defined until relatively late stages of the disease when clinical characteristics listed in Table 3 become prominent. Therefore, defining the molecular events occurring prior to pathological and clinical manifestations could enable detection of NE/SC at earlier stages rather than at the end of the disease spectrum. To this end, we will summarize and discuss the molecular events underlying NE/SC development in the remaining sections of the review.

## 4. Loss of *TP53* and *RB1* Is a Backbone of NE/SC Development

At the genomic level, there is a substantial overlap of genomic alterations between CRPC adenocarcinoma and NE/SC [30]. Next-generation sequencing of patient-derived specimens revealed that a loss of *RB1* (a gene encoding RB protein) and *TP53* (a gene encoding p53 protein) is a common denominator of genomic aberrations in NE/SC [1,30], although *TP53/RB1* loss also occurs in CRPC adenocarcinoma at lower frequency [1]. Given a dynamic phenotypic switch from adenocarcinoma to NE/SC, it is critical to determine the extent to which *TP53*/*RB1* loss is responsible for NE/SC development. In fact, a large proportion of molecular events underlying NE/SC development may be explained by ramifications of *TP53* and *RB1* inactivation as we discuss below. Figure 1 illustrates the molecular events underlying development of NE/SC to be discussed in the remaining sections of the review. 

### 4.1. Activation and Dysregulation of E2F1 Cistrome Caused by RB1 Loss

In normal non-proliferating cells, hypophosphorylated RB inhibits the activity of E2Fs (including E2F1 upregulated in NE/SC [1]) through direct binding [31]. Upon mitogenic stimulation, the sequential hyperphosphorylation of RB by activated cyclin-dependent kinases results in the loss of RB function, followed by release of E2Fs from RB, promoting the expression of E2Fs target genes necessary for cell cycle progression and DNA synthesis [31]. In PCa, *RB1* loss is rare in primary disease [32,33,34], but ADT may select for tumor loci with low *RB1* activity to bypass cell-cycle blockage imposed by ADT [35].

Accordingly, in NE/SC where *RB1* is often inactivated, E2F1 activity is elevated [1]. In general, E2Fs including E2F1 engage in cell-fate decision makings in many cell types, and their activation results in stem cell expansion, inhibition of differentiation and altered lineage choices upon differentiation [36]. In PCa, *RB1* loss not only induces E2F1 binding to canonical E2F1 targets, but also reprograms E2F1 transcriptional activity by inducing E2F1 binding to non-canonical E2F1 targets [37].

### 4.2. Consequences of Activated and Dysregulated E2F1 Cistrome

Overactivation and dysregulation of E2F1 cistrome with expanded binding capacity contributes to NE/SC development by inducing the expression of several key transcription factors and epigenetic modifiers. The first example of dysregulated E2F1 cistrome is the reprogramming transcription factor SOX2 which is a known E2F1 target gene [38,39]. SOX2 is a critical factor to maintain pluripotency as well as a lineage specifier for neural lineage [40]. In PCa, SOX2 elevated by loss of *TP53* and *RB1* has been reported to drive lineage plasticity promoting NE/SC development [41]. This SOX2-driven lineage switching accompanies global hypomethylation of both histone H3 at lysine 4 and histone H3 at lysine 9, which may be mediated by LSD1 (lysine-specific demethylase 1) [42].

NE/SC is known to possess epigenetic profiles including DNA methylation [30,43] and histone modification [44] distinct from those of CRPC adenocarcinoma. A key player is EZH2 (enhancer of zeste homolog 2) whose expression is also elevated in NE/SC [1]. EZH2 is another direct downstream target of *RB1*-E2F pathway [45], and considered to be a master regulator of epigenetic rewiring in NE/SC [46].EZH2 is a catalytic subunit of the Polycomb repressive complex 2 (PRC2), and PRC2 epigenetically represses genes required for differentiation to maintain the pluripotent state [47]. In support of EZH2 as a master regulator in NE/SC, EZH2 inhibition has been shown to prevent NE/SC development in PCa cell lines [48]. Furthermore, Phase I clinical trial of EZH2 inhibitor (Tazemetostat) in combination with AR-targeted drugs against AVPC is currently underway (NCT04179864). In addition, the efficacy of Lirametostat (CPI-1205, a selective inhibitor of EZH2) in combination with enzalutamide or abiraterone is currently being evaluated in metastatic CRPC cohort (NCT03480646) [49], though its potential benefit to NE/SC patients has yet to be evaluated.

As another example of epigenetic regulators altered by *TP53/RB1* loss, NE/SC expresses a higher level of methylation maintenance enzyme called DNMT1 (DNA methyltransferase 1) [30]. Here again, p53 and RB play a pivotal role in DNMT1 regulation by transcriptionally suppressing DNMT1 [50], which explains DNMT1 elevation in NE/SC that lost the function of *TP53* and *RB1*. Moreover, EZH2 physically interacts with DNMTs by serving a recruitment platform and this interaction is required for CpG methylation of EZH2-target promoters [51], and trimethylation of H3K27 (histone H3 Lys27) carried out by EZH2 is necessary for *de novo* DNMTs-mediated DNA methylation [52]. These findings highlight the intricate interplay among these epigenetic modifiers downstream of *RB1*-E2F pathway.

Elevated E2F1 activity by *RB1* loss also leads to AR upregulation as E2F1 is recruited to the AR regulatory locus and induces AR expression [32]. Although AR transcriptional activity is low in fully grown NE/SC [1], a substantial fraction (35%) of CRPC with combined biallelic loss of *RB1* and *TP53* was classified as AR-positive adenocarcinomas without neuroendocrine features [53]. Thus, E2F1-mediated AR elevation may drive progression of prostate tumors with AR dependency prior to eventual loss of this dependency during NE/SC development. Further supporting the significant impacts of *RB1* loss and the resultant E2F1 cistrome dysregulation, E2F1 has been reported to bind to promoter regions of NE markers such as CGA, SYP and NSE upon *RB1* knockdown [54].

A recent study focused on E2F1 as a therapeutic target of NE/SC. Since E2F1 itself is not currently directly targetable, BRD4 that cooperates with E2F1 to activate NE/SC lineage plasticity program has been targeted with BET bromodomain inhibitors (BETi) [55]. BETi showed therapeutic effects not only in preclinical models, but also in some NE/SC patients treated with BETi ZEN-3694 (NCT02711956) [55], highlighting the importance of E2F1 in NE/SC development. Taken together, several critical regulators downstream of E2F1 orchestrate NE/SC development.

### 4.3. The Role of TP53 Loss in NE/SC Development

What is the role of *TP53* inactivation during NE/SC development? It is well established that *TP53* mutations, often associated with loss of function, are more frequently detected in CRPC (prior to NE/SC) than *RB1* loss [33]. In a preclinical mouse model, inactivation of *Rb* alone only led to PIN (prostatic intraepithelial neoplasia) formation and concurrent deletion of *Tp53* was required for the development of metastatic carcinoma with neuroendocrine features [56], suggesting *TP53* loss may be a prerequisite for functions mediated by *RB1* loss. This can be explained by the function of *TP53* as a cellular gatekeeper, in which stress signals including loss of tumor suppressors induce several downstream events such as cell cycle arrest and apoptosis, and not surprisingly, *RB1* loss is one of these *TP53* activating events [57]. More specifically, E2F1 accumulation caused by *RB1* loss subsequently activates the ARF/Mdm2/p53 pathway, leading to induction of apoptosis [58]. Therefore, *TP53* inactivation may be required for CRPC cells to fully exploit the aforementioned benefits of dysregulated E2F1 cistrome during the development of NE/SC signified by *RB1* loss.

In addition to *RB1* loss, oncogene activation is also a *TP53* inducer [57]. In the context of NE/SC, for instance, when oncogenic *PEG10* (another E2F1 target gene that promotes invasion and proliferation of NE/SC; to be detailed in the next section) is overexpressed in LNCaP harboring WT *TP53*, *TP53* expression is induced and growth promotion does not occur, whereas cell growth increase once *TP53* is knocked down [59]. Another example of TP53-promoting oncogene is MYCN (also to be detailed in the next section). MYCN amplification has been reported to activate TP53 and induce subsequent apoptosis in neuroblastoma [60]. Therefore, in NE/SC in which MYCN expression is often elevated [61], loss of TP53 may be required for MYCN to fully exert its oncogenic effects.

To summarize, although *TP53/RB1* loss is not unique to NE/SC, current data support the nomination of *TP53/RB1* loss as the common genomic denominator necessary for NE/SC development.

## 5. Additional Alterations Required for NE/SC Development Are Often Associated with ADT

Although it seems almost unquestionable that loss of *TP53* and *RB1* is necessary for NE/SC, next-generation sequencing data of matched DNA and RNA show that a substantial fraction (35%) of CRPC with combined biallelic loss of *TP53* and *RB1* was classified as AR-positive adenocarcinomas without neuroendocrine features [53]. Furthermore, double knockout of *TP53* and *RB1* in LNCaP PCa cell lines did not induce the expression of NE-associated genes [53]. Therefore, additional genetic or epigenetic alterations that potentially tilt the cell fate towards full-fledged NE/SC may be required.

LTL331 PDX (patient-derived xenograft) is an elegant model that demonstrates a transition from adenocarcinoma (LTL331) to NE/SC (LTL331R). Interestingly, adenocarcinoma LTL331 already harbors heterozygous copy loss at the *RB1* and *TP53* gene loci, and the hemizygous loss-of-function mutations in the remaining alleles, meaning that both genes are already severely compromised. Yet, LTL331 does not transform to NE/SC until castration of the host, suggesting that castration (aka ADT) plays an important role in NE/SC development in this model [59], which is also consistent with clinical observations that longer duration of ADT increases NE/SC incidence [62]. The LTL331 model has been used to define key molecular events as a consequence of ADT that drives the transition from adenocarcinoma to NE/SC. For example, *PEG10* (Paternally Expressed 10, one of E2F1 target genes) was identified as a key molecule. In LTL331, AR inhibits the expression of *PEG10* by a direct binding to *PEG10* promoter, whereas castration results in elevated PEG10 expression, promoting invasion and proliferation of NE/SC [59].

Another important molecule, REST (RE1-silencing transcription factor), is also under the control of AR. REST suppresses the expression of neuronal genes in non-neuron-related tissues. Regarding REST regulation by AR, AR inhibits REST degradation by inhibiting a REST ubiquitin ligase [63]. Thus, following castration, REST expression is downregulated, and loss of REST in turn triggers the expression of neuron-specific genes during NE/SC development [64]. Although other study suggested that REST loss only induces the expression of a limited set of NE/SC-associated genes and is not sufficient to fully transform CRPC to NE/SC [65], it is highly likely that REST loss plays a part in NE/SC development.

N-Myc (encoded by *MYCN*), a well-known oncogenic transcription factor, is significantly elevated in NE/SC [61], and is capable of transforming human prostate epithelial cells to NE/SC in vitro [66]. Notably, ADT reprograms N-Myc cistrome in the direction towards neural lineage. In the presence of androgen, N-Myc binds to regulatory sequences associated with AR binding upstream of AR target genes. Upon androgen withdrawal, N-Myc is redirected towards promoters of neural-lineage genes leading to transcriptional activation of these genes [67]. Additionally, E2F1 cistome reprogramming discussed in the previous section may be involved in this N-Myc activity augmentation. Myc target is one of pathways enriched upon *RB1* loss, implicating dysregulated E2F1 cistrome in N-Myc upregulation [37]. Although direct activation of N-Myc by E2F1 is yet to be demonstrated in PCa and further studies are needed, E2Fs have been reported to promote N-Myc expression by binding to *MYCN* promoter in other cancer types [68], raising the possibility that N-Myc elevation, triggered by dysregulated E2F1 cistrome, may contribute to NE/SC development.

N-Myc activity is also intricately associated with AURKA (Aurora kinase A) in NE/SC. AURKA and N-Myc are concurrently overexpressed in NE/SC [61]. Overexpression of AURKA results in aneuploidy and tumorigenesis [69]. In NE/SC, AURKA exerts its oncogenic effects through a feedforward loop with N-Myc. N-Myc enhances the expression of AURKA and AURKA increases N-Myc stability by inhibiting N-Myc-targeting ubiquitin ligase [70]. AURKA inhibition appears to be effective against a certain subset of N-Myc/AURKA-high NE/SC [66]. Interestingly, AR enhances the expression of AURKA [71], giving rise to a question regarding how AURKA is upregulated in AR-indifferent NE/SC. The feedforward interaction between AURKA and N-Myc suggests that N-Myc can drive AURKA expression independent of AR in NE/SC.

ADT also induces the expression of proneural transcription factor ASCL1 (Achaete-Scute Complex-Like 1) [72] and BRN2 (encoded by *POU3F2*) [73]. ASCL1 and BRN2 promote neurogenesis and are two of three transcription factors that are required to convert non-neural somatic cells into neurons [74]. ASCL1 together with NKX2-1 (also elevated by castration in LTL331 model) have been reported to induce reprogramming of FOXA1 cistrome [75]. Specifically, in adenocarcinoma, FOXA1 binds to regulatory elements of prostate-lineage genes such as *KLK3, HOXB13*, and *NKX3-1*. FOXA1 dramatically changes its cistrome in NE/SC and binds to regulatory elements of NE/SC-associated genes. This FOXA1 cistrome reprogramming was reproduced by ectopic expression of ASCL1 and NKX2-1 [75]. BRN2 is highly expressed in NE/SC, and is directly repressed by AR which binds to the androgen response element (ARE) of BRN2 [73]. BRN2 induces the expression of NE markers and confers aggressive phenotypes to NE/SC [73].

A recent study has shed light on the pleiotropic functions of MUC1-C (an oncogenic C-terminal transmembrane subunit of Mucin1) in this intricate interplay among aforementioned proneural and pluripotent transcription factors. As opposed to AR which represses BRN2 expression, MUC1-C activates BRN2 expression by directly binding to BRN2 promoter in collaboration with MYC. Furthermore, silencing MUC1-C decreases expression of ASCL1 and the four OSKM pluripotency factors (OCT4, SOX2, KLF4 and MYC) [76]. In line with the importance of AR suppression in NED, AR suppresses MUC1-C transcription by directly binding to ARE in MUC1-C promoter region [77].

Collectively, the findings summarized above support the general concept that while dual inactivation of *TP53* and *RB1* is the foundation of NE/SC development, molecular events triggered by ADT are required for the completion of NE/SC development. The transition from adenocarcinoma to NE/SC therefore may involve NE/SC-associated key transcription factors and cistrome reprogramming that further tilt the cell fate towards neuroendocrine lineage.

## 6. Histological Classification of PCa Disease Continuum from Adenocarcinoma to NE/SC and Associated Molecular Events

In general, the transition from adenocarcinoma to NE/SC can be explained by molecular events occurring under the selective pressure of potent AR-targeting agents in the context of *TP53/RB1* loss. However, the transition process should be considered a temporally and spatially complex process of tumor evolution. There, histological and molecular variants during NE/SC development should be expected. Recently, Labrecque et al. proposed classifying disease continuum between CRPC adenocarcinoma and NE/SC into the following five categories: ARPC (adenocarcinomas with AR and PSA), AR-low PCa (weak AR and PSA, and negative for NE markers), amphicrine PCa (coexpress AR, PSA, and NE markers), double-negative PCa (DNPC, negative for AR, PSA, and NE markers) and NE/SC [65].

Given the spatial and temporal heterogeneity of the disease, it is challenging to precisely delineate temporal molecular events described in the previous sections during transition from adenocarcinoma to NE/SC. However, the LTL331 model which accurately recapitulates the clinical course of a patient from whom LTL331 was derived gives us some clues regarding the key molecule drivers. Table 4 summarizes the gene expression changes of molecules discussed in this review during NE/SC development in LTL331 model [59]. It takes 6-8 months for LTL331 (adenocarcinoma) to become LTL331R (NE/SC) after castration, and although the term “double-negative” was not used in this Akamatsu et al.’s paper, their IHC data showed that LTL331 becomes negative for AR, PSA and chromogranin A at post-castration 2–3-month timepoint, which recapitulates DNPC [59]. Therefore, one can surmise that in the context of *TP53/RB1* loss (in LTL331), several additional molecular events that depend on AR suppression evolve to drive the development of NE/SC. Histological and molecular intermediates in the process of this transition have been evaluated more carefully in this model. With regards to histological intermediate phases between adenocarcinoma and NE/SC, a set of genes associated with squamous cancers was elevated in a subset of AR-low PCa and DNPC, implying the existence of a transition state to squamous CRPC. However, a determinant to switch cell fate to squamous lineage is yet to be determined [65]. Bluemn et al. particularly investigated DNPC, and found elevated FGF and MAPK signaling pathway in DNPC, enabling bypass of AR dependence [78]. In addition, DNPC has been shown to acquire mesenchymal and stem-like traits through the activation of PRC1 (Polycomb Repressor Complex 1), and PRC1 promotes metastasis of DNPC [79].

Last but not least, the West Coast SU2C-PCF Dream Team recently proposed a morphologically distinct subtype of PCa called intermediate atypical prostate cancer (IAC) that shows features of both adenocarcinoma and NE/SC (NCT02432001) [80]. IAC showed a gene expression signature that is intermediate to adenocarcinoma and NE/SC [80]. How IAC is related to the other categories of CRPC proposed in Labrecque et al. (ARPC, AR-low PCa, amphicrine PCa and DNPC [65]) remains to be determined.

## 7. Early Detection of NE/SC

How can we harness the molecular understanding of NE/SC and make use of the key molecular drivers for early detection of NE/SC? Several lines of evidence show that aberrations in *TP53, RB1* and *PTEN* are associated with poor clinical outcomes. For example, deleterious variants of *TP53* and *RB1* were associated with poorer outcomes both in localized and metastatic diseases [34]. PCa with gene expression profile associated with *TP53/RB1* loss showed worse overall survival (OS) and poor treatment response to AR-targeted therapies [53]. Not surprisingly, these aberrations in *RB1, TP53* and *PTEN* in aggressive diseases overlap with molecular features of clinically defined AVPC [28]. In other words, detection of these aberrations early may help to predict the emergence of NE/SC. Although frequency is lower in comparison with advanced diseases, *TP53, RB1* and *PTEN* alterations can be readily detected in primary localized prostate cancer [33,34], and alterations in *TP53, RB1* and *PTEN* are associated with increased risk of relapse in localized disease [34]. Other biomarkers that predict the emergence of NE/SC at the treatment-naïve stage are *AURKA* and *MYCN* amplifications [81]. It has been reported that a large proportion of patients who develop NE/SC at later stages of the disease already carry concurrent *AURKA* and *MYCN* amplifications in their primary PCa in comparison with a random and unselected PCa cohort [81]. Thus, when those alterations are detected in primary PCa, patients should be closely monitored, and intensive therapeutic intervention and a possibility of NE/SC emergence should be considered.

Early detection of NE/SC may be particularly relevant in the setting of systemic therapies. In this setting, it is often not feasible to acquire tissue biopsies. Cell-free DNA (cfDNA) testing can be a feasible option during generally long-term PCa follow-up periods. McNair et al. has shown that copy number alterations in *RB1* and RB pathway genes can be readily traced by cfDNA testing [37]. Given the contribution of *RB1* loss and consequent E2F1 cistrome dysregulation to NE/SC development as described in the previous section, this point-of-care RB testing may lead to clinical benefit [37]. To further explore the potential of cfDNA testing in NE/SC early diagnosis, Beltran et al. employed the combined panels of NE/SC-associated genomic and epigenomic alterations to identify NE/SC patients [82]. In their study, there was a patient whose cfDNA profile showed NE/SC characteristics prior to actual clinical manifestations of NE/SC liver metastasis, suggesting that cfDNA testing is useful to capture NE/SC-related molecular events earlier than clinical manifestations [82].

Finally, little has been done from NE/SC prevention perspective. As androgen withdrawal plays a pivotal role in NE/SC development [59,62], androgen supplementation may delay the onset of NE/SC. Bipolar androgen therapy (BAT) recently came under the spotlight and is showing promising results so far in terms of re-sensitization to enzalutamide [83]. It would be informative to identify patients at high risk for NE/SC (e.g., those carrying *TP53, RB1* and *PTEN* alteration or *AURKA/MYCN* amplification in their primary PCa or biopsies at later stages) and compare the long-term prevalence of NE/SC in BAT-treated patients versus untreated patients.

Together, it is possible to predict the emergence of NE/SC to a certain extent based on primary PCa tissue specimens as well as liquid biopsies. Further development in these areas will be necessary to drive efforts in early detection of NE/SC and facilitate development of rationalized treatment strategies.

## 8. Conclusions

In this review, we summarized clinical and pathological characteristics of NE/SC and reviewed molecular events underlying transition from adenocarcinoma to NE/SC with a special focus on key drivers including inactivation of *TP53/RB1* and other drivers emerging upon androgen withdrawal. Some molecular definitions may be readily utilized in the clinical settings to complement existing pathological and clinical definitions. Although tissue- and liquid-biopsy based testing for the purpose of predicting NE/SC is still at the exploratory stage, it is a promising area of research and development that will enable early detection and precision therapy for a lethal form of prostate cancer trending to rise in frequency due to early use of potent AR-targeting agents.

## Figures and Tables

**Figure 1 ijms-22-12742-f001:**
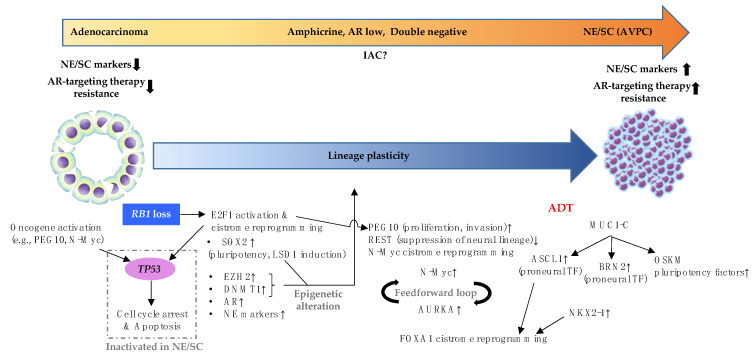
Schematic presentation of molecular events during NE/SC development. Abbreviations: IAC-intermediate atypical prostate cancer, AVPC—aggressive variant prostate cancer, OSKM—*OCT4*, *SOX2*, *KLF4* and *MYC*.

**Table 3 ijms-22-12742-t003:** Clinical features of AVPC proposed by Aparicio et al. [4].

CRPC with at Least One of the Following (Patients with Small-Cell Prostate Carcinoma on Histologic Evaluation Were Not Required to Have Castration-Resistant Disease):
Histologic evidence of small-cell prostate carcinoma (pure or mixed). Exclusively visceral metastases. Radiographically predominant lytic bone metastases by plain x-ray or CT scan. Bulky (≥5 cm) lymphadenopathy or bulky (≥5 cm) high-grade (Gleason ≥ 8) tumor mass in prostate/pelvis. Low PSA (≤10 ng/mL) at initial presentation (before ADT or at symptomatic progression in the castrate setting) plus high volume (≥20) bone metastases. Presence of neuroendocrine markers on histology (positive staining of chromogranin A or synaptophysin) or in serum (abnormal high serum levels for chromogranin A or GRP) at initial diagnosis or at progression. Plus any of the following in the absence of other causes: A. elevated serum LDH (≥2 × IULN) B. malignant hypercalcemia C. elevated serum CEA (≥2 × IULN). Short interval (≤6 months) to androgen-independent progression following the initiation of hormonal therapy with or without the presence of neuroendocrine markers.

Abbreviations: GRP—gastrin-releasing peptide, IULN—Institutional Upper Limit of Normal, LDH—Lactate dehydrogenase, CEA—carcinoembryonic antigen. Modified from [4].

**Table 4 ijms-22-12742-t004:** Relative expression of key molecules during NED based on RNA sequencing data of the LTL331 model. Extracted from Akamatsu et al. Supplementary Table S2 [59]. Normalized mRNA read counts were deivided by read counts at pre Cx1.

	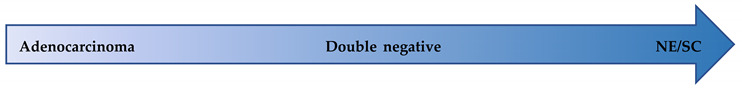
Gene Name	LTL331 Pre Cx1	LTL331 Pre Cx2	LTL331 Post Cx 8wk	LTL331 Post Cx 12wk	LTL331 NE/SC 1	LTL331 NE/SC 2
*ASCL1*	1.00	0.55	0.33	3.82	2.52	1.52
*AURKA*	1.00	1.06	0.43	0.26	1.75	2.20
*DNMT1*	1.00	1.43	1.43	1.19	2.95	2.93
*E2F1*	1.00	0.94	0.63	0.36	3.37	3.79
*FOXA1*	1.00	1.11	0.96	0.81	0.12	0.13
*MUC1*	1.00	0.39	0.24	0.33	2.18	1.14
*MYCN*	1.00	1.16	2.27	2.34	6.82	6.84
*NKX2-1*	1.00	5.17	17.71	11.02	5.13	13.95
*PEG10*	1.00	7.31	18.68	19.82	370.77	702.05
*POU3F2* (BRN2)	1.00	0.54	0.70	1.45	146.84	210.52
*REST*	1.00	1.28	1.57	1.76	0.02	0.01
*SOX2*	1.00	2.87	1.22	12.25	6350.99	4626.95

Abbreviations: Cx—Castration.

## Data Availability

Not applicable.

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
