# Peer review of "Delineating the Molecular Events Underlying Development of Prostate Cancer Variants with Neuroendocrine/Small Cell Carcinoma Characteristics"

_ijms, 2021, doi:10.3390/ijms222312742_

Round 1

Reviewer 1 Report

This review by Kanayama and Luo compiles extensively the molecular events underlying transition from CRPC adenocarcinoma to neuroendocrine prostate cancer with a special focus on key drivers including inactivation of RB1 /TP53 and other drivers with a special emphasis on RB1/E2F alteration. As the authors mentioned it is a promising area of research given the fact the treatment induced NEPC comprises around 20% of the prostate cancer.

There are some out of focus description and comments on the manuscript.

  1. The abstract – lines 14-16 describing PSAM may not necessary, given the fact the manuscript did not consider any other later therapeutic agents like radioligands, immunotherapy etc.
  2. Table 1 should be recreated by including WHO and Epstein classification
  3. P53 loss of function in NEPC deserves more explanation
  4. Reference list is little extensive – please shorten the list.

  minor comments

  1. check line 84 for RB or RB1
  2. line 91 – prostate lineage markers – does it restrict with AR and PSA – or the PSA restricts with luminal markers?

Author Response

Response to reviewers:

We sincerely thank the reviewers for their time and their positive comments. We have revised the original manuscript to address comments by reviewer # 1.

Reviewer #1

This review by Kanayama and Luo compiles extensively the molecular events underlying transition from CRPC adenocarcinoma to neuroendocrine prostate cancer with a special focus on key drivers including inactivation of RB1 /TP53 and other drivers with a special emphasis on RB1/E2F alteration. As the authors mentioned it is a promising area of research given the fact the treatment induced NEPC comprises around 20% of the prostate cancer.

There are some out of focus description and comments on the manuscript.

Thank you for evaluating our work. We sincerely appreciate your feedback. We have addressed all concerns as detailed below. Revisions made to the manuscript are indicated in red.

  1. The abstract – lines 14-16 describing PSMA may not necessary, given the fact the manuscript did not consider any other later therapeutic agents like radioligands, immunotherapy etc.

We agree with the reviewer that this review is not therapy focused. Thus, we eliminated the following sentence from the abstract “In addition, PSMA targeting radioligands are expected to further improve the prognosis of patients with advanced prostate cancer.”

  1. Table 1 should be recreated by including WHO and Epstein classification.

We agree with the reviewer that including WHO classification further clarifies existing criteria. Therefore, we added WHO classification to Table1.

  1. P53 loss of function in NEPC deserves more explanation

We agree with the reviewer that TP53 loss should be explained in detail in the context of NE/SC. However, we also note the vast literature on TP53 that may be out of the scope of this review. Thus, we added the following relevant contexts to the manuscript and Figure 1:

“Another example of TP53-promoting oncogene is MYCN (also to be detailed in the next section). MYCN amplification has been reported to activate TP53 and induce subsequent apoptosis in neuroblastoma [58]. Therefore, in NE/SC in which MYCN expression is often elevated [59], loss of TP53 may be required for MYCN to fully exert its oncogenic effects.”

  1. Reference list is little extensive – please shorten the list.

We appreciate reviewer’s comment, and have revised the references. However, we refrained from shortening the list substantially as doing so may compromise the clarity of this review. We looked at some published examples of this special issue “Molecular Advances in Prostate Cancer”, and one review cited more than 200 papers (we cited 81 papers).

  minor comments

  1. check line 84 for RB or RB1

We have changed it to RB1.

  1. line 91 – prostate lineage markers – does it restrict with AR and PSA – or the PSA restricts with luminal markers?

            We appreciate reviewer’s comment and realize this is probably a confusing statement. We have revised the sentence as follows:

NEPCa generally does not express markers generally specific to prostate cells (e.g., PSA), but characteristically expresses markers of neural lineage such as chromogranin A, synaptophysin, and CD56 (Table 2)”.

Reviewer 2 Report

This manuscript is a review about clinical and pathological characteristics of NE/SC and molecular events underlying transition from CRPC adenocarcinoma to NE/SC. As PCa is one of the most frequent and studied cancer, especially in men, this paper puts light on a clue aspect for therapy management. In my opinion, apart from minor English check, this manuscript can be accepted in the present form.

Author Response

Reviewer #2

This manuscript is a review about clinical and pathological characteristics of NE/SC and molecular events underlying transition from CRPC adenocarcinoma to NE/SC. As PCa is one of the most frequent and studied cancer, especially in men, this paper puts light on a clue aspect for therapy management. In my opinion, apart from minor English check, this manuscript can be accepted in the present form.

Thank you for evaluating our work. We sincerely appreciate your feedback.